# The Role of Uniform Meropenem Usage in *Acinetobacter baumannii* Clone Replacement

**DOI:** 10.3390/antibiotics10020127

**Published:** 2021-01-29

**Authors:** Bence Balázs, Zoltán Tóth, Fruzsina Nagy, Renátó Kovács, Hajnalka Tóth, József Bálint Nagy, Ákos Tóth, Krisztina Szarka, László Majoros, Gábor Kardos

**Affiliations:** 1Department of Medical Microbiology, Faculty of Medicine, University of Debrecen, 4032 Debrecen, Hungary; balazs.bence@med.unideb.hu (B.B.); toth.zoltan@med.unideb.hu (Z.T.); nagy.fruzsina@med.unideb.hu (F.N.); kovacs.renato@med.unideb.hu (R.K.); toth.hajnalka13@gmail.com (H.T.); nagy.jozsefb93@gmail.com (J.B.N.); szkrisz@med.unideb.hu (K.S.); major@med.unideb.hu (L.M.); 2Doctoral School of Pharmaceutical Sciences, University of Debrecen, 4032 Debrecen, Hungary; 3National Public Health Center, 1097 Budapest, Hungary; toth.akos@nnk.gov.hu

**Keywords:** carbapenem resistant *Acinetobacter baumannii*, blaOXA-23-like-carbapenemase, blaOXA-40-like carbapenemase, clonal dynamics, strain replacement, multidrug resistance, changepoint analysis

## Abstract

The dominant carbapenem resistant *Acinetobacter baumannii* harboring *bla_OXA-23-like_* carbapenemase was replaced by *bla_OXA-40-like_* carriers in a Hungarian tertiary-care center with high meropenem but relatively low imipenem use. We hypothesized that alterations in antibiotic consumption may have contributed to this switch. Our workgroup previous study examined the relation between resistance spiral and the antibiotic consumption, and the results suggest that the antibiotic usage provoked the increasing resistance in case of *A. baumannii.* We aimed at measuring the activity of imipenem and meropenem to compare the selection pressure exerted by the different carbapenems in time-kill assays. Strain replacement was confirmed by whole genome sequencing, core-genome multilocus sequence typing (cgMLST), and resistome analysis. Based on results of the time-kill assays, we found a significant difference between two different sequence-types (STs) in case of meropenem, but not in case of imipenem susceptibility. The newly emerged ST636 and ST492 had increased resistance level against meropenem compared to the previously dominant ST2 and ST49. On the other hand, the imipenem and colistin resistance profiles were similar. These results suggest, that the uniform meropenem usage may have contributed to *A. baumannii* strain replacement in our setting.

## 1. Introduction

*Acinetobacter baumannii* became an important nosocomial pathogen, especially in ventilator-associated pneumonia, wound, urinary tract, and bloodstream infections [1]. Three international clones (ICs) have been determined; IC I (includes ST1; ST7; ST8; ST19; ST20); IC II (includes ST2; ST45; ST47; ST59); IC III (includes ST3; ST13; ST14). In Europe, IC I and IC II cause the vast majority of infections and IC II was reported in most cases [2,3]. *A. baumannii* harbors chromosomally-encoded AmpC-type cephalosporinase and a chromosomal OXA-51-like oxacillinase [4,5,6]. The mentioned cephalosporinase together with porin mutations can be associated with low-level carbapenem resistance [7]. The chromosomally encoded OXA-51-like oxacillinase enzymes can also be linked to carbapenem resistance, especially in case of their overproduction when their expression is facilitated by upstream *ISAba* insertion sequences. Carbapenem resistance caused by these mechanisms appeared in the 1980s first, but this mechanism was soon replaced by plasmid-borne carbapenem-hydrolyzing class D β-lactamases (CHDLs) more efficient in conveying carbapenem resistance, i.e., OXA-23-like and OXA-58-like, later OXA24/40-like [4,6,8,9,10,11]. The *bla_OXA-23-like_* gene family has been found in *A. baumannii* in 1995 and detected worldwide in hospital outbreaks [12,13,14,15,16,17]. The *bla_OXA24/40-like_* genes were found in clinical isolates from northern Spain in 2002 [2]. When purified, all OXA carbapenemases show relatively weak activity against carbapenems, significant clinical resistance is due to the concerted action of the oxacillinases with promoting effects of *ISAbas*, overexpression of efflux pumps, and variations of different outer membrane proteins (ompA, carO, oprD, and omp33).

Our working group first investigated the molecular epidemiology of carbapenem resistant *A. baumannii* (*CRAb*) and the effect of carbapenem consumption on its distribution published in a previous study, where six clones were identified of which five were carbapenem resistant due to *bla_OXA-23-like_* genes (78.1% of all tested isolates), while the *bla_OXA-24/40-like_* gene was found in two sporadic isolates (1.2%) [17]. The present study investigates the changes in the occurrence of carbapenem resistance genes between 2012 and 2017 in relation to carbapenem usage patterns and characterizes representative isolates from the former (2010) study and from this period to uncover potential mechanisms in the background of strain dynamics.

## 2. Results

### 2.1. Antibiotic Consumption

Carbapenem consumption increased more or less steadily between 2005 and 2018 (Mann–Kendall tau = 0.684, *p* < 0.0001; Figure 1), meropenem usage increased four times between the two endpoints; mean usage was 0.64 defined daily dose per 100 occupied bed-days (DDD/100 OBDs) in 2007 while 2.69 DDD/100 OBDs in 2017. Imipenem consumption was similar in 2007 (mean usage 0.48 DDD/100 OBDs), but it did not change dramatically during the study period (mean usage 0.80 DDD/100 OBDs in 2017), as shown in Figure 1.

### 2.2. Resistance Phenotype

Antibiotic resistance was over 75% to all tested drugs except colistin (Appendix A). Carbapenem resistance steadily increased during the study (Mann–Kendall tau = 0.610, *p* < 0.0001). Over 90% of the isolates have been *CRAb*. Resistance to ciprofloxacin and gentamicin remained very high throughout the study, while amikacin and tobramycin resistance decreased in 2016–17 significantly (*p* < 0.0001). Colistin resistance was sporadic throughout.

### 2.3. Prevalence of the Resistance Genes

Prevalence of the aminoglycoside resistance genes decreased during the examination period (Appendix A). The dominant *aph(3′)-VIa* phosphotransferase gene prevalence decreased to 6.2% from 90.6% (*p* < 0.0001). The *aac(6′)-Ib* acetyl-transferase and the *armA* methyl-transferase show a similar pattern (*p* < 0.0001).

All of the isolates contained the *bla_OXA-51-like_* genes proving that all investigated isolates were *A. baumannii* (Appendix A). The dominant carbapenemase gene family between 2010 and 2014 was the *bla_OXA-23-like_* (Appendix A), but occurrence of the gene decreased significantly in 2016 and 2017 (*p* < 0.001). The prevalence of *bla_OXA-24/40-like_* carbapenemases showed an opposite trend, while in the beginning of the examination period the gene family was almost totally absent, its prevalence increased significantly in 2016 (*p* < 0.001), and it was even higher in 2017 (*p* < 0.001).

### 2.4. Whole Genome Sequencing (WGS)

In 2010, *bla_OXA-23-like_* carbapenemases proved to be *bla_OXA-23_* in all sequenced strains representing all major pulsotypes, which belonged to ST1, ST2, and ST45. Only two isolates of the same minor pulsotype carried the *bla_OXA-24/40-like_* gene [17]; one of which proved to be ST79 and a *bla_OXA-24_* carrier by WGS. All representative isolates from 2017 harboring only *bla_OXA-23-like_* acquired carbapenemase gene belonged to ST2. Representative isolates of the more abundant *bla_OXA-24/40-like_* carriers (75.0% in 2016 and 92.3% in 2017 of the total isolates), in turn, belonged to ST636 and ST492, all harboring *bla_OXA-72_* or, in case of some ST636 isolates, *bla_OXA-72_* together with *bla_OXA-23_* (Table 1). The genetic environment of the *bla_OXA-23-like_* gene in these latter isolates was identical to that found in ST45 strains from 2010, also including the gene *aph(3′)-Via* [17]. No other types of carbapenemase genes were found in the sequenced genomes.

The WGS confirmed acquisition of the *bla_OXA-23_* carbapenemase at least two times in 2010, in case of ST1 and ST45. ST636 isolates were genetically highly uniform, distance based on allele presence is ≤4 alleles, while ST2 is more diverse, three of four sequenced isolates belonged to a subgroup distinct from the 2010 ST2 isolates by 20 alleles (Figure 2).

### 2.5. Time-Kill Assays

The examined isolates were resistant against meropenem and imipenem in clinical aspects, the minimum inhibitor concentrations (MICs) were >32 mg/L, except the only *bla_OXA-51-like_* (*bla_OXA-69_* or the *bla_OXA-66_*) harboring isolates (MIC = 2 mg/L in both cases). Against *bla_OXA-23_* carriers (ST2), meropenem showed a concentration-dependent bactericidal effect (Figure 3a) between 128 and 1024 mg/L (mean k = 0.48; 0.13–0.59), while *bla_OXA-72_* carriers (ST636; ST492) were not killed at 256 mg/L (mean k = −0.27; −0.49–−0.01), only 512–1024 mg/L (mean k = 0.14; 0.00–0.28) were bactericidal, moreover, only a bacteriostatic effect was detected at 1024 mg/L against isolates harboring both genes (mean k = −0.14; 0.00–−0.25). Meropenem was bactericidal at 32 mg/L against those isolates, which harbored only the *bla_OXA-69_* gene (chromosomally encoded *bla_OXA-51-like_* gene) (mean k = 0.08; 0.05–0.10). Imipenem showed concentration-independent killing between 128 and 1024 mg/L similar against *bla_OXA-72_* (mean k = 0.17; 0.05–0.31) or *bla_OXA-23_* (mean k = 0.36; 0.19–0.57) carrier strains, against isolates carrying *bla_OXA-23_* and *bla_OXA-72_* simultaneously imipenem was bactericidal at 256 mg/L (mean k = 0.43; 0.16–0.71) (Figure 3b). Similarly, meropenem was uniformly ineffective at lower (16–32 mg/L) concentrations, while imipenem was slightly more effective against *bla_OXA-72_* carriers at 64 mg/L (data not shown). The effect of colistin was isolate rather than strain dependent (Figure 3c).

### 2.6. Change-Point Analysis

*CRAb* incidence exhibited nine changepoints (Figure 4). Out of the antibiotic consumption, time-series changepoints corresponding with time-series of carbapenem use were found only in case of *CRAb* incidence (February 2009 vs. June 2009; October 2014 vs. December 2014).

## 3. Discussion

The role of the selective pressure exerted by antibiotics as a provoker of resistance is well known. However, the response to selection pressure may be versatile and evolves over time. Hospital ecology of multiresistant bacteria is characterized by emergence and receding of different bacterial strains, is shaped by virulence, survival ability, and transmissibility of these strains and is fundamentally influenced by the selection pressure exerted by antibiotic use; shown to provoke spread and persistence of resistant strains by a number of studies. For example, the increasing amikacin usage provoked appearance of amikacin resistant but tobramycin sensitive *Acinetobacter*, while reducing amikacin consumption decreased the number of the resistant isolates [18]. Similar findings were reported between the intravenous fluoroquinolone usage and the fluoroquinolone-resistant *A. baumannii* [19].

In this study, the patterns of carbapenemase gene carriage and results of the WGS strongly suggest a strain switch, i.e., the newly appeared ST636 and ST492 *CRAb* strains carrying *bla_OXA-72_* carbapenemase gene replaced the ST2 and ST45 *bla_OXA-23_* carriers during the period 2014–2015. Similar emergence and spread of *bla_OXA-24/40-like_* carrier strains have been reported from several countries [20,21,22,23,24,25,26,27]. The strain replacement reported hereby is in line with the well-documented evolution of carbapenem resistance in *A. baumannii* from overexpressed *bla_OXA-51-like_* enzymes in sporadic isolates through *bla_OXA-23-like_* and *bla_OXA-58-like_* enzymes with potent carbapenem hydrolyzing activity carried by pandemic clones to emerging clones carrying *bla_OXA-24/40-like_* enzymes with even more efficient hydrolytic activity. This evolution towards more efficient hydrolysis of carbapenems is linked to increasing use of carbapenems in many countries, as documented earlier in this setting [17] and elsewhere [28,29,30,31]. This relationship is the inherent part of the Gram-negative resistance spiral, as carbapenem usage increases in response to the spread of extended spectrum β-lactamase (ESBL) producing Enterobacterales in our setting [32]. The link between the strain switch and carbapenem use is further supported by the coincident changepoints.

Sequenced ST636 isolates were highly similar, corresponding to clonal spread of these isolates within this setting, which may be an advent of the emergence of a new high-risk clone. Some isolates within this clone acquired another carbapenemase (*bla_OXA-23_*) most probably from ST45, one of the earlier resident clones, suggesting that ST636 is currently evolving in the setting.

Investigating the pattern of antibiotic consumption as a potential driver of this change, notable differences were found between killing kinetics of imipenem and meropenem. While meropenem killed the novel strains ST492 and especially ST636 markedly less efficiently at higher concentrations, the effect of imipenem was comparable in case of all examined strains. As monthly meropenem use exceeds imipenem use 5–10-fold, the difference in meropenem resistance levels may have provided the selective advantage fueling the emergence of *bla_OXA-72_* producing strains, especially ST636 with exceedingly high meropenem resistance. This assumption may similarly be applied in other settings; in Colombia Villalobos et al. reported a high prevalence of *CRAb* together with strong preference for meropenem over imipenem in Colombia [33].

This is in line with the well-documented process that the more resistant strain is expected to outcompete the less resistant provided that the selection pressure is high enough to exceed the fitness cost of resistance [34], i.e., if it has a fitness advantage in a situation with high antibiotic pressure. This is elegantly demonstrated in case of trimethoprim and *Escherichia coli* by Baym et al. in an ingenious experimental setting [35]. It was also shown that transmission between patients is also strongly linked to drug use [36,37].

This principal precept is also applicable to small differences in fitness in the long run. Though the less efficient killing found in the present case may provide only a small advantage at clinically attainable carbapenem concentrations, such small competitive advantages may be important in selection between resistant strains as well as in shaping competitive interactions. For instance, in case of subclones of *E. coli* ST131, the success of pandemic H30Rx clones with very high fluoroquinolone resistance over other resistant subclones was linked to fluoroquinolone use [38] or methicillin-resistant *Staphylococcus aureus* (*MRSA*) strain dynamics was driven by drug use patterns [39]. Thus, it may be assumed that the problem leading to and maintaining the problem of *CRAb* lies not only in overuse, but also in imbalanced consumption of carbapenems, i.e., the preference of meropenem over imipenem as empirical as well as targeted therapy. This is further supported by the acquisition of another potent carbapenemase by the already highly resistant ST636 and the extremely high level of meropenem resistance (but not of imipenem resistance) in these isolates, pointing out that strain evolution in this setting is driven towards even higher level of carbapenem resistance. Thus, even strains with clinically relevant high-level resistance may be outcompeted by more resistant novel clones under high selection pressure and thus attaining higher resistance levels may be evolutionarily advantageous even for highly resistant strains with significant clinical resistance.

Antibiotic stewardship efforts generally aim at reducing consumption of a drug family, considering individual drugs with the same spectra within this family equivalent. The present study draws attention to the fact that related compounds may not be equal, and one may favor spread of certain resistant strains and/or fuel evolution of higher resistance levels in already resistant strains. The other way around, versatile use of individual compounds within a drug group may oppose spread of resistance by decreasing uniform selection pressure. When total drug restriction is not feasible, such an approach may be advantageous.

## 4. Materials and Methods

### 4.1. Clinical Isolates

Samples were identified by matrix-assisted laser desorption/ionization time-of-flight (MALDI-TOF) Biotyper (Bruker, Daltonics). Nonduplicate *A. baumannii* isolates were collected between 2012 and 2017 (*n* = 74 in 2012; *n* = 118 in 2013; *n* = 128 in 2014; *n* = 136 in 2016 and *n* = 65 in 2017) at the Clinical Centre of the University of Debrecen, Hungary. Isolates from 2010–2011 have been characterized in an earlier work [17]. Susceptibility to amikacin, tobramycin, gentamicin, colistin, imipenem, meropenem, and ciprofloxacin were determined using the European Committee on Antimicrobial Susceptibility Testing (EUCAST) guidelines. Changes in susceptibility were analyzed by means of chi-square test.

### 4.2. Antibiotic Consumption and Changepoint Analysis

We collected monthly antibiotic consumption data between 2005 and 2017 in defined daily dose (DDD) per 100 occupied bed days (OBDs). Trend analysis was performed using a Mann–Kendall test in R statistical environment (Kendall package, MannKendall test). Time-series of *CRAb* incidence density per 100 bed-days and of consumption of different antibiotic groups were subjected to changepoint analysis in R statistical environment (changepoint package; changepoints in mean). To identify drug groups possibly involved, the changepoints of the *CRAb* incidence density was matched to changepoints of the different drug use series (coamoxiclav, third generation cephalosporins, carbapenems, fluoroquinolones, aminoglycosides, and colistin) to find temporally close changepoint patterns.

### 4.3. Resistance Genes

The carbapenemase genes *bla_OXA-23-like_*, *bla_OXA-24/40-like_*, and *bla_OXA-51-like_* and insertion sequences *ISAba-1* were sought for using the methods of Woodford et al. and Turton et al. [8,9,40,41,42,43]. Occurrence of the aminoglycoside resistance genes *aac(6′)-Ib*; *aph(3′)-Ia* and *armA* were investigated using the methods of Frana et al., Vila et al., and Bogaerts et al. [41,42,43].

### 4.4. Whole Genome Sequencing

For whole genome sequencing, isolates representing all pulsotypes (A1; A2; B; C1; C2; D and those representing unique isolates) of the 2010/11 study (*n* = 9) were chosen. In this study, the isolates from 2017 (*n* = 14) were selected based on their differences in the resistance phenotype (i.e., susceptible to all investigated antibiotics, amikacin resistant but tobramycin susceptible; susceptible only to gentamicin and colistin, susceptible only to ciprofloxacin and colistin, and resistant against all investigated antibiotics except colistin).

Genomic DNA for all isolates was extracted by DNeasy UltraClean Microbial Kit (Qiagen, Hilden, Germany). Libraries were created using Nextera DNA Flex library preparation kit (Illumina, San Diego, CA, USA) and sequenced on MiSeq platform (Illumina) using MiSeq Reagent Kit v2 (300 cycles) (Illumina). The resulting FASTQ files were quality trimmed and then de novo assembled using the Velvet assembler integrated in Ridom SeqSphere+ software (Ridom GmbH, Münster, Germany). The cgMLST analysis was performed using SeqSphere+ software (Ridom) according to the ‘*A. baumannii* cgMLST’ version 1.0 scheme. Acquired antibiotic resistance genes were identified with ResFinder v3.9 (cge.cbs.dtu.dk/services/ResFinder/) and Comprehensive Antibiotic Resistance Database (CARD) v3.1.0 (card.mcmaster.ca/analyze). The raw sequence reads were uploaded to NCBI BioProject database (BioProject ID: PRJNA671692), accession numbers of the isolates are shown in Table 1.

### 4.5. Time-Kill Assay

In time-kill assays the effect of meropenem, imipenem, and colistin were investigated against 12 carbapenem resistant *A. baumannii* isolates representing all major STs (ST2; ST636, ST1, ST492). The starting inoculum was 10^6^ CFU/mL, the drug concentrations were 16, 128–1024 mg/L in case of carbapenems, and 2–32 mg/L in case of colistin in Mueller-Hinton broth medium (Lab M Limited, Heywood, UK). In a smaller subset of strains (*n* = 4) the meropenem and imipenem time-kill assays were also performed at lower concentrations (16 and 32 mg/L) as well. All tubes were plated at 0, 2, 4, 6, 8, 10, 12, and 24 h to solid Mueller-Hinton media (Lab M Limited, Heywood, UK). The colonies were counted after 24 h of incubation at 37 °C. Killing rates (k) were determined: *N_t_*  =  *N*_0_  ×  e^−*kt*^, where *N_t_* is the number of viable bacteria colonies at time *t*, *N*_0_ is the number of viable bacteria colonies in the initial inoculum, *k* is the killing rate, and *t* is the incubation time. Negative and positive *k* values indicate growth and killing, respectively. Bactericidal activity was defined as a 99.9% reduction of the starting inoculum. All tests were performed at least twice.

## Figures and Tables

**Figure 1 antibiotics-10-00127-f001:**
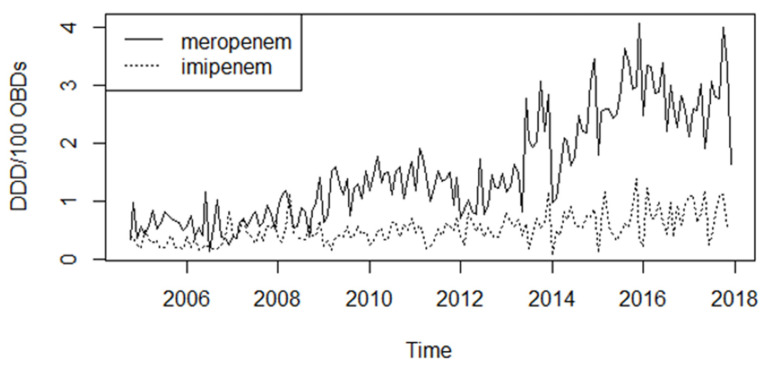
Changes in meropenem and imipenem consumption between 2004 and 2018 in tertiary-case in defined daily doses per 100 occupied bed-days (DDD/100 OBDs).

**Figure 2 antibiotics-10-00127-f002:**
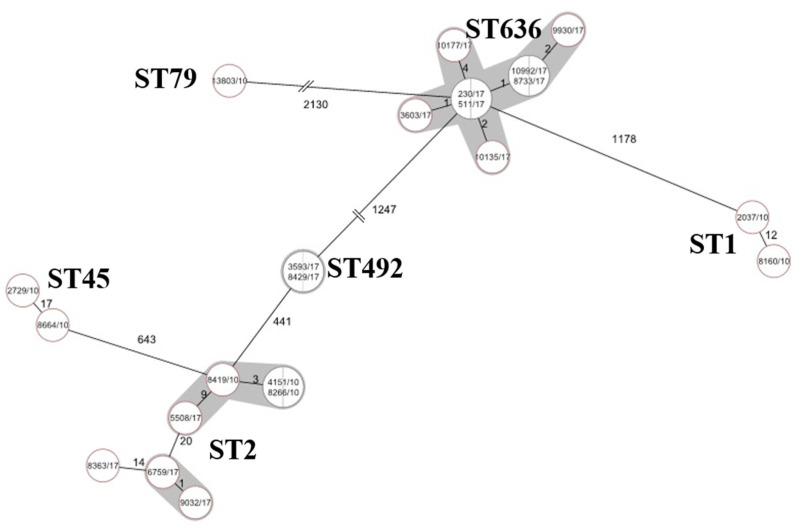
Phylogenic tree of sequenced *A. baumannii* isolates based on core-genome multilocus sequence typing (cgMLST). Minimum spanning tree based on cgMLST allelic profiles of 22 *A. baumannii* isolates. Each circle represents an allelic profile based on sequence analysis of 2390 cgMLST target genes. The numbers on the connecting lines illustrate the numbers of target genes with different alleles. Closely related genotypes (<10 alleles difference) are shaded.

**Figure 3 antibiotics-10-00127-f003:**
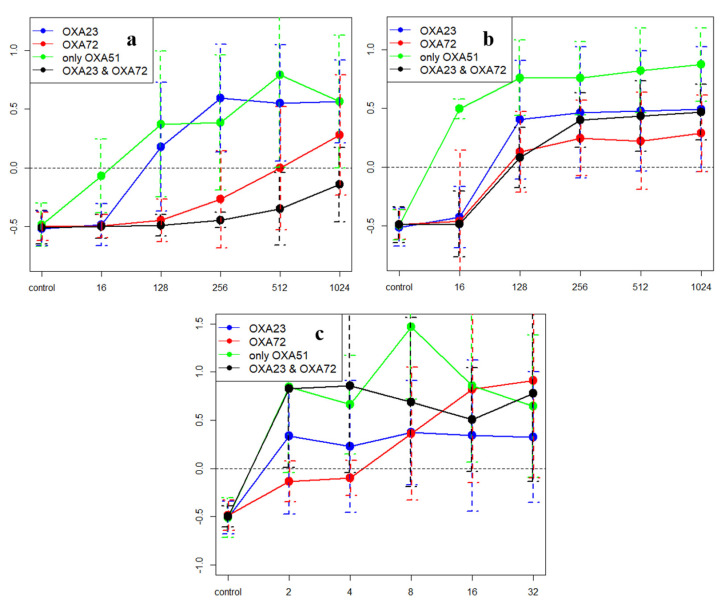
Result of the time-kill assays, X axis shows the drug concentration, Y axis shows the mean k-values. Values above 0 (dashed line) indicate killing effect (see text for details). (**a**) Summary of the results of the time-kill assays against meropenem. (**b**) Summary of the results of the time-kill assay against imipenem. (**c**) Summary of the results of the time-kill assay against colistin.

**Figure 4 antibiotics-10-00127-f004:**
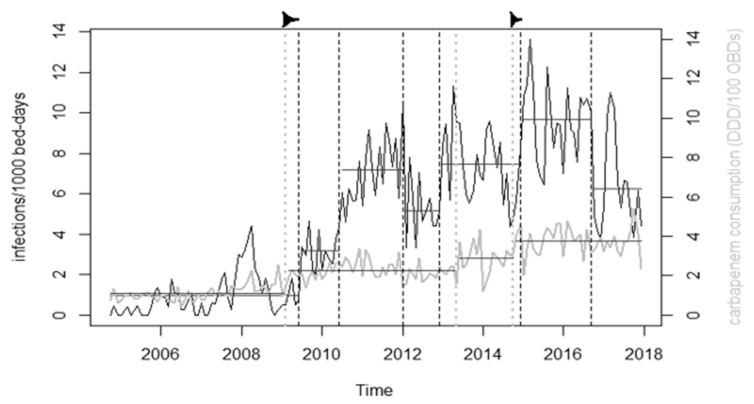
Results of the changepoint analysis. Grey line shows carbapenem consumption, black line shows incidence density of infections by *CRAb*. Dashed vertical lines mark the changepoints found, horizontal lines mark averages of sections of the time-series between changepoints. The two arrows mark the correspondences between changepoints in carbapenem consumption and incidence.

**Table 1 antibiotics-10-00127-t001:** Result of the whole genome sequencing of selected isolates and relationship between *Acinetobacter baumannii* sequence types and the antibiotic resistance genes carried. Ac.No. = accession number; ST = sequence type; CHDLs = class D β-lactamases; Bl = β-lactams; Qui = quinolones; Agl = aminoglycosides; Sul = sulphonamides; Mac = macrolides; Phe = phenicols; Tet = tetracyclines.

Isolate	Sample	Year	ST	CHDLs	Resistance Genes						
					Bl	Qui	Agl	Sul	Mac	Phe	Tet
4151 Ac.No.: SRX9354489	Blood	2010	2	blaOXA-23 blaOXA-69	blaADC-25	aac(6′)-Ib-cr	aac(6′)-Ib3 aadA1 aph(3′’)-Ib aph(3′)-Ia aph(3′)-VIa aph(6)-Id armA	sul1	mph(E) msr(E)	catB8	tet(B)
8266 Ac.No.: SRX9354496	Bronchial	2010	2	blaOXA-23 blaOXA-66	blaADC-25	aac(6′)-Ib-cr	aac(6′)-Ib3 aadA1 aph(3′’)-Ib aph(3′)-Ia aph(3′)-VIa aph(6)-Id armA	sul1	mph(E) msr(E)	catb8	tet(B)
8419 Ac.No.: SRX9354497	Bronchial	2010	2	blaOXA-23 blaOXA-66	blaADC-25	aac(6′)-Ib-cr	aac(6′)-Ib3 aadA1 aph(3′’)-Ib aph(3′)-Ia aph(3′)-VIa aph(6)-Id armA	sul1	mph(E) msr(E)	catB8	tet(B)
2738 Ac.No.: SRX9354478	Throat	2010	2	blaOXA-23 blaOXA-66	blaADC-25	aac(6′)-Ib-cr	aac(6′)-Ib3 aadA1 aph(3′’)-Ib aph(3′)-Ia aph(3′)-VIa aph(6)-Id armA	sul1	mph(E) msr(E)	catB8	tet(B)
2037 Ac.No.: SRX9354477	Abscess	2010	1	blaOXA-23 blaOXA-69	blaADC-25		aac(3)-Ia aadA1 aph(3′)-VIa	sul1			
8160 Ac.No.: SRX9354495	Urine	2010	1	blaOXA-69	blaADC-25		aac(3)-Ia aadA1 aph(3′)-VIa	sul1			
8664 Ac. No.: SRX9354494	Canule	2010	45	blaOXA-23 blaOXA-66	blaADC-25		aph(3′)-Ia aph(3′)-VIa	sul1		catA1	
2729 Ac.No.: SRX9354493	Bronchial	2010	45	blaOXA-23 blaOXA-66	blaADC-25	aac(6′)-Ib-cr	aac(6′)-Ib3 aph(3′)-Ia	sul1		catA1	tet(A)
13803 Ac.No.: SRX9354498	Blood	2010	79	blaOXA-24 blaOXA-65	blaADC-25 blaTEM-1B		aac(3)-IIa aac(6′)-Ian aph(3′’)-Ib aph(3′)-VIa aph(6)-Id armA	sul2		cmlB1	
5508 Ac.No.: SRX9354499	Bronchial	2017	2	blaOXA-23 blaOXA-66	blaOXA-23	aac(6′)-Ib-cr	aac(6′)-Ib3 aadA1 aph(3′’)-Ib aph(3′)-Ia aph(6)-Id armA	sul1	mph(E) msr(E)	catB8	tet(B)
6759 Ac.No.: SRX9354479	Sputum	2017	2	blaOXA-23 blaOXA-66	blaADC-25	aac(6′)-Ib-cr	aac(6′)-Ib3 aadA1 aph(3′’)-Ib aph(3′)-Ia aph(3′)-VIa aph(6)-Id armA	sul1	mph(E) msr(E)	catB8	tet(B)
8363 Ac.No.: SRX9354482	Bronchial	2017	2	blaOXA-23 blaOXA-66	blaADC-25	aac(6′)-Ib-cr	aac(6′)-Ib3 aadA1 aph(3′’)-Ib aph(3′)-Ia aph(6)-Id armA	sul1	mph(E) msr(E)	catB8	tet(B)
9032 Ac.No.: SRX9354481	Sputum	2017	2	blaOXA-23 blaOXA-66	blaADC-25	aac(6′)-Ib-cr	aac(6′)-Ib3 aadA1 aph(3′’)-Ib aph(3′)-Ia aph(3′)-VIa aph(6)-Id armA	sul1	mph(E) msr(E)	catB8	tet(B)
230 Ac.No.: SRX9354485	Bronchial	2017	636	blaOXA-72 blaOXA-66			aac(3)-Ia aadA1 aph(3′)-Ia	sul1		catA1	
511 Ac.No.: SRX9354486	Bronchial	2017	636	blaOXA-72 blaOXA-66			aac(3)-Ia aadA1 aph(3′)-Ia	sul1		catA1	
3603 Ac.No.: SRX9354487	Bronchial	2017	636	blaOXA-72 blaOXA-23 blaOXA-66	blaADC-25		aac(3)-Ia aadA1 aph(3′)-Ia aph(3′)-VIa	sul1		catA1	
9930 Ac.No.: SRX9354492	Bronchial	2017	636	blaOXA-72 blaOXA-23 blaOXA-66	blaADC-25		aac(3)-Ia aadA1 aph(3′)-Ia aph(3′)-VIa	sul1			
10135 Ac.No.: SRX9354488	Nasal	2017	636	blaOXA-72 blaOXA-66	blaADC-25		aac(3)-Ia aadA1	sul1		catA1	
10177 Ac.No.: SRX9354490	Throat	2017	636	blaOXA-72 blaOXA-66	blaADC-25		aac(3)-Ia aadA1	sul1		catA1	
10992 Ac.No.: SRX9354483	Bronchial	2017	636	blaOXA-72 blaOXA-23 blaOXA-66	blaADC-25		aac(3)-Ia aadA1 aph(3′)-Ia aph(3′)-VIa	sul1			
8733 Ac.No.: SRX9354491	Drain	2017	636	blaOXA-72 blaOXA-66	blaADC-25		aac(3)-Ia aadA1 aph(3′)-Ia	sul1			
3593 Ac.No.: SRX9354480	Bronchial	2017	492	blaOXA-72 blaOXA-66	blaADC-25		aadA2 aph(3′’)-Ib aph(6)-IdarmA	sul1 sul2	mph(E) msr(E)		tet(B)
8429 Ac.No.: SRX9354484	Trachea	2017	492	blaOXA-72 blaOXA-66	blaADC-25		aadA2 aph(3′’)-Ib aph(6)-IdarmA	sul1 sul2	mph(E) msr(E)		tet(B)

## Data Availability

The genomes sequenced in this study are available in the GenBank under the bioproject ID PRJNA671692.

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
