# Peer review of "The Role of Uniform Meropenem Usage in Acinetobacter baumannii Clone Replacement"

_antibiotics, 2021, doi:10.3390/antibiotics10020127_

Round 1

Reviewer 1 Report

This article by Bence Balazs et al. describe very interesting data on the role of meropenem usage in Acinetobacter baumannii clone replacement. 

Minor revision is needed before acceptance.

Global :

  • Bacterial names, genes, "i.e.", "et al." etc. need to be written in italic.
  • Acronyms have to be fully explained before their first use.

Part 2.5.: Data could also be summarized by a table to ease the understanding of the very interesting data provided by the TKC. Maybe authors could complete the table S1 and consider it in the main manuscript?

Part 4.1. : please give the country for Dubrecen (Hungary ?)

Part 4.4. :Lots of details are lacking and have to be given about the selection criteria for the "representative isolates of pulsotypes", the method for library production before WGS and the bioinformatic method used for their analyses.

Supplementary figures/tables have to be excluded from the manuscript (as they could been downloaded separately).

Suppress part 5.

Author Response

Answer to comments of Reviewer 1

We thank the Reviewer for the helpful comments and suggestions.

Comment 1: Bacterial names, genes, "i.e.", "et al." etc. need to be written in italic.

Response: We italicized the names throughout.

Comment 2: Acronyms have to be fully explained before their first use.

Response: Acronyms were defined at the first mention.

Comment 3: Part 2.5.: Data could also be summarized by a table to ease the understanding of the very interesting data provided by the TKC. Maybe authors could complete the table S1 and consider it in the main manuscript?

Response: As suggested, we converted Table S1 into Table 1 adding it to the main manuscript. However, when trying to present the time-kill results in table form, this resulted in two different tables (one for carbapenems and one for colistin), which were highly redundant with Figure 3. Therefore, we decided to keep the text format, as this offers the opportunity to compare and explain different groups of isolates more clearly and to highlight important differences.

Comment 4: Part 4.1. : please give the country for Debrecen (Hungary ?)

Response: We completed the text.

Comment 5: Part 4.4. :Lots of details are lacking and have to be given about the selection criteria for the "representative isolates of pulsotypes", the method for library production before WGS and the bioinformatic method used for their analyses

Response: We added the data required.

Comment 6: Supplementary figures/tables have to be excluded from the manuscript (as they could been downloaded separately).

Response: We corrected the mistakes.

Comment 7: Suppress part 5.

Response: We removed the leftover subtitle.

We hope the changes improved the manuscript.

Yours sincerely,

Gábor Kardos

corresponding author

Reviewer 2 Report

The action of OXAs as carbapenemases is an issue of great concern in the clinic. The increased use of β-lactam-based drugs is a leading reason for the proliferation and diversification of β-lactamases. In this manuscript, Balazs et al. investigate the change in OXA genes' occurrence in relation to carbapenem usage patterns. Besides, the authors characterize the representative isolates by whole genome sequencing. This work would be of broad interest to readers of the antibiotics journal. However, some items need to be addressed before publication.

  1. Joshi et al. (2017) report the co-existence of blaOXA-23 and blaNDM-1 genes in Acinetobacter baumannii. Chen et al. (2018) detect the co-existence of blaNDM-1 and OXA-58. The authors must detect the presence of the blaNDM-like, blaIMP-like, and blaVIM-like genes. It can be done by either PCR or searching the whole genomes of isolates.
  2. The authors need to improve the quality of Figure 3. The authors are suggested to replot the figure and use a different color for each object. Besides, the drug concentrations were 16, 64-1024mg/L in the case of carbapenems and 2-32mg/L in the case of colistin in the time-kill assay (line 262). Figure 3A/B miss the 64 mg/L concentration.
  3. It would be nice to replace the blaOXA-40-like with blaOXA-24/40-like in Figure S3.
  4. The authors should remove the conclusion section (there is no content). 

Author Response

Answer to comments of Reviewer 2

The authors thank the Reviewer for the comments and suggestions to which we answer as follows.

Comment 1: Joshi et al. (2017) report the co-existence of blaOXA-23 and blaNDM-1 genes in Acinetobacter baumannii. Chen et al. (2018) detect the co-existence of blaNDM-1 and OXA-58. The authors must detect the presence of the blaNDM-like, blaIMP-like, and blaVIM-like genes. It can be done by either PCR or searching the whole genomes of isolates.

Response: As shown in Table 3 (Table S1 in the previous version), no carbapenem-hydrolysing lactamases were found based on the whole genome sequencing apart from the blaOXA genes using ResFinder v3.9 (cge.cbs.dtu.dk/services/ResFinder/) and Comprehensive Antibiotic Resistance Database (CARD) v3.1.0 (card.mcmaster.ca/analyze). A sentence stating this was added to the results.

Comment 2: The authors need to improve the quality of Figure 3. The authors are suggested to replot the figure and use a different color for each object. Besides, the drug concentrations were 16, 64-1024mg/L in the case of carbapenems and 2-32mg/L in the case of colistin in the time-kill assay (line 262). Figure 3A/B miss the 64 mg/L concentration.

Response: We replotted the mentioned Figure 3, and corrected the mistake about the carbapenem concentrations. In the time-kill assays the concentrations normally used were 16; 128-1024 mg/L in case of carbapenems. The concentration 64 mg/L was incorrectly mentioned in the text, which was amended. This mistake arose from the fact that for some isolates, we also investigated the concentrations 32 and 64 mg/L as stated in the text. As these concentrations did not differ from the concentration 16 mg/L in case of the few isolates where they were tested, these were not used for k value calculation and plotting. The new plots show the correct concentrations.

We also redrawed the plots in colour to make them better readable.

Comment 3: It would be nice to replace the blaOXA-40-like with blaOXA-24/40-like in Figure S3.

Response: We made the changes requested.

Comment 4: The authors should remove the conclusion section (there is no content).

Response: We corrected the mistake.

We hope the changes improved the manuscript.

Yours sincerely,

Gábor Kardos

corresponding author